

# On the uncertainty associated with detecting global and local mean sea level drifts on Sentinel-3A and Sentinel-3B altimetry missions

Rémi Jugier[1], Michaël Ablain[1], Robin Fraudeau[1], Adrien Guerou[2], Pierre Féménias[3]
[1]MAGELLIUM, Ramonville Saint-Agne, 31520, France
[2]CLS Collecte Localisation Satellites, Ramonville Saint-Agne, 31520, France
[3]ESA/ESRIN, Frascaty, Italy
*Correspondence to*: Rémi Jugier (remi.jugier@magellium.fr)
**Abstract.** An instrumental drift in the Point Target Response (PTR) parameters has been detected on the Copernicus Sentinel-
3A (S3A) altimetry mission. It could have an impact on sea level rise of a few tenths of mm yr$^{-1}$. In order to accurately evaluate
this drift, a method for detecting global and local mean sea level relative drifts between two altimetry missions is implemented.
Associated uncertainties are also accurately calculated thanks to a detailed error budget analysis. A drift on both S3A and S3B
GMSL is detected with values significantly higher than expected. For S3A, the relative GMSL drift detected is 1.0 mm yr$^{-1}$
with Jason-3 and 1.3 mm yr$^{-1}$ with SARAL/AltiKa. For S3B, the relative GMSL drift detected is -2.2 mm yr$^{-1}$ with
SARAL/AltiKa and -3.4 mm yr$^{-1}$ with Jason-3. The drift detected at global level does not show detectable regional variations
above the uncertainty level of the proposed method. The investigations led by the altimeter experts can now explain the origin
of this drift for S3A, while it is still under investigation for S3B. The ability of the implemented method to detect a sea level
drift with respect to the length of the common period is also analysed. We find that the maximum detectable sea-level drift
over a 5 years period is 0.3 mm yr$^{-1}$ at the global scale, and 1.5 mm yr$^{-1}$ at local scales (2400 km). However, these levels of
uncertainty do not meet the sea-level stability requirements for climate change studies.
## 1. Introduction
Sea level is one of the key indicators of climate change, integrating the changes of mass in the ocean from glaciers and polar
ice melt, changes in temperature of the ocean from the excess heat entering the earth system (Meyssignac et al., 2019, von
Schuckmann et al., 2020), as well as changes in land water storage (Chambers et al., 2017). As such, the Global Mean Sea
Level (GMSL) has been defined by the Global Climate Observing System (GCOS) as an Essential Climate Variable (ECV),
and the GMSL rise is a widely accepted indicator for the rate at which the climate is changing.
Since 1993, the GMSL indicator has been calculated on a continual basis from four reference altimetry missions
(TOPEX/Poseidon (T/P), Jason-1, Jason-2 and Jason-3), all on the same orbit. The GMSL time series of each altimeter have
been accurately linked together through inter-calibration during the tandem phases (Zawadzki and Ablain, 2016) : T/P--Jason-
1, Jason-1--Jason-2, then Jason-3--Jason-2. The satellites follow each other very closely throughout these phases and therefore
measure the water surface height with nearly identical oceanic and atmospheric conditions. The description of the errors, and





the uncertainties on the long-term stability of the sea level estimate for these products were provided by Ablain et al. (2019)
and Prandi et al. (2021) for the global and local scales respectively. Over the whole altimetry period (January 1993-December
2020), the GMSL shows a significant rise of $+3.48 \pm 0.35$ mm yr$^{-1}$. At the local scale, the sea level rise distribution ranges
between 0 and 6 mm yr$^{-1}$, with uncertainties ranging from $\pm 0.8$ to $\pm 1.2$ mm yr$^{-1}$, indicating that the sea level is rising everywhere
over the globe. Recent studies also showed that sea level is accelerating at $0.12 \pm 0.07$ mm yr$^{-2}$ at the global scale (Ablain et
al., 2019) and ranges between -1 mm yr$^{-2}$ and +1 mm yr$^{-2}$ at the regional scale (Prandi et al., 2021). The Sentinel-6 Michael
Freilich (S6-MF) mission, recently launched in November 2020 on the same historical T/P-Jason orbit, will allow the GSML
time series to be extended once the current validation phase is completed (early 2022).
Several other altimetry missions (e.g. ERS-1, ERS-2, Envisat, Cryosat-2, SARAL-AltiKa) have also been launched from 1991
onwards, all in different orbits at lower altitudes and with lower revisit rate (e.g. 35 days for Envisat). Although these missions
were not designed to be as stable as T/P, Jasons and S6-MF, their data is nevertheless very relevant to improve and verify the
long-term stability of the climate altimeter record. On the one hand, data from these complementary missions combined with
data from the reference climate missions can generate value-added products with higher spatio-temporal resolution and better
global coverage towards the poles (e.g. sea level products from CMEMS (Taburet et al., 2019) and C3S (Legeais et al., 2021)).
On the other hand, cross-comparison of complementary and reference altimetry missions over the same period allows for
verification of the coherence of sea level measurements between these missions and possibly detection of relative drifts
between them (e.g. Envisat GMSL drift (Ollivier et al., 2012)).
More recently, the Sentinel-3A (S3A) and Sentinel-3B (S3B) altimetry missions, developed in the framework of the European
Space Agency (ESA) Copernicus program, were launched in February 2016 and April 2018 to provide sea level measurements
for Copernicus operational services (e.g. CMEMS, C3S). They complete the existing constellation of altimeter satellites based
on Jason-2, Jason-3, SARAL/AltiKa and Jason-3, to which must be added the Cryosat-2 and HY-2A/2B missions. S3A and
S3B are equipped with a SAR/Doppler altimeter instead of a conventional altimeter like the climate reference missions. This
new altimeter has a much better along-track resolution, and its measurements are very accurate. However, this mission is not
aimed primarily towards climate studies and MSL stability over time. An unexpected behaviour of the S3A altimeter was
indeed pointed out by Poisson et al. (2019): the drift of the point target response (PTR) parameters was higher than expected,
with a direct impact on the GMSL trend of about 0.3 mm yr$^{-1}$.
Our main motivation for this study is to verify whether this instrumental drift of the S3A and S3B missions can be detected by
comparing the GMSLs of the different altimetry missions available over the same period. The verification of the stability of
S3A and S3B data with the new SAR mode is an important issue as well, to anticipate the stability of the S6-MF mission,
which also uses this technology and which will soon be the reference mission to calculate the GMSL indicator.
Therefore, this study aims, in the first place, to accurately estimate the relative GMSL drift of S3A, Jason-3 and SaRAL/Altika
missions over all the S3A period (from March 2016 to August 2021), and the relative GMSL drift of S3B, S3A, Jason-3 and
SaRAL/Altika missions over all the S3B period (from June 2018 to August 2021). Since the comparison periods are short (5
years for S3A and 3 years for S3B), high levels of uncertainties can be expected on the GMSL difference trend estimates. An



important question is whether the small expected GMSL drift on S3A (0.3 mm yr$^{-1}$ from PTR parameter drift, see Poisson et
al., 2019) can be detected on such short periods. Hence, a main objective of this study is to provide the uncertainty estimates
of the GMSL drift calculation with their confidence interval levels. Using this uncertainty calculation, we will be able, on the
first hand, to affirm whether a drift of the S3A or S3B GMSL is detected, and on the other hand, we will be able to show in a
general way the capacity of the cross-calibration methods to detect GMSL drifts according to the length of the period. In the
context of climate change study, this information is very important to continue to improve on the GMSL time series in order
to meet the more stringent sea level stability requirements provided (e.g. 0.1 mm yr$^{-1}$ for the GMSL trend from Meyssignac,

72 2019).

Since the potential GMSL drift detected on S3A and S3B could have a regional signature, we also propose to extend the
detection of sea level drift to local scales. Similarly to the global scale, the objective is to estimate the ability of the cross-
calibration method to detect a sea level drift at local scales by taking into account the length of the temporal series and the size
of the spatial scale from a few hundred to a few thousand km. This will allow us to evaluate the regional drift on S3A and S3B
and determine what level of drift can be detected with this type of approach.
In the following paper, we first focus on the description of the data used and the methods applied to calculate global and local
mean sea level drifts. A great attention is given to the mathematical formalism applied to calculate the uncertainties. Then, we
describe and analyse the relative mean sea level obtained between the different altimetry missions, accounting for the
uncertainty estimates and discussing the sensitivity of the obtained results.
**2. Altimeter Data**
Since the S3A launch in February 2016, Jason-3 and SARAL/AltiKa have continuously provided high-quality sea level
measurements, as reported in the validation reports of both missions (see (Roinard and Michaud, 2020) and (Jettou and
Rousseau, 2020)). Furthermore, Jason-3 has also been the reference mission since October 2016 for computing the GMSL
indicator on AVISO (https://www.aviso.altimetry.fr/msl). These two missions are therefore selected in this study to perform
cross-comparisons with S3A from July 2016 onwards (the first months after the S3A mission launch between February and
July 2016 were used for calibration purposes and are therefore not suited for GMSL measurement). For the same reasons,
Jason-3 and SARAL/AltiKa are selected to perform cross-comparison with S3B from December 2018 onwards, as well as S3A
which also covers the entire S3B period. Other altimetry missions partially cover the S3A or S3B periods like Jason-2, HY-
2A, HY-2B, and Cryosat-2. Among these missions, only Jason-2 could be chosen because of its very good stability, however
the end of life of the mission in October 2019 reduces the interest to use these data for cross-comparisons with S3A, Jason-3
and SARAL/AltiKa.
The altimeter products used are the non-time critical (NTC) along-track Level-2+ (L2P) products from the Copernicus
Altimetry Marine service under the CNES responsibility for Jason-3 and SARAL/AltiKa, and Eumetsat responsibility for S3A



and S3B. These products contain the along-track sea level anomaly (SLA, see Eq. (1)) calculated after applying a validation
process fully described in the product handbook of each altimeter mission.

$$SLA = Orbit - Range - \sum_i Correction_i - MeanSeaSurface \tag{1}$$


Furthermore, the geophysical corrections applied in L2P products for the SLA calculation are homogenised for all the missions
allowing us to reduce the discrepancies between each altimetry mission.
The wet tropospheric correction from on-board radiometers is an important source of GMSL drift (see (Ablain et al., 2019)).
However, in this study, we choose to focus on altimeter induced drift. We therefore use the same wet tropospheric correction
for all missions, derived from the operational ECMWF model (distributed in the L2P products). This effectively eliminates
uncertainties from the wet tropospheric correction when calculating GMSL differences, allowing for a more accurate
assessment of the altimeter drift.

**3. Method**

**3.1. Calculation of GMSL differences**

The most straightforward way to compute GMSL differences (noted $\Delta GMSL$ hereafter) between two altimetry missions is to
compute SLA grids ($MSLA(lon, lat, t)$), 1 degree along the latitudinal axis and 3 degrees along the longitudinal axis) on
common time periods of 10 days, and then compute the difference between the SLA grids. The 10-day period corresponds
approximately to the repeatability (i.e. duration of a cycle) of the reference climate missions (TOPEX/Poseidon, Jasons (1,2,3),
S6-MF). We then compute a global weighted mean of the grid differences, weighted by the ocean surface of each cell, in
identical fashion to the GMSL AVISO indicator (https://www.aviso.altimetry.fr/msl/). All grid cells above 66° of latitude and
under -66° are also eliminated in order to homogenize the spatial coverage of the different missions, restricted by Jason-3. It
is calculated by weighting ($w_i (lon, lat)$ ) each box according to its latitude and its area covering the ocean, in order to give
less significance to boxes at high latitudes which cover a smaller area and to boxes that overlap land.

$$\Delta GMSL(t) = \frac{\sum_{lon,lat} w_i(lon, lat) * \Delta MSLA(lon, lat, t)}{\sum_{lon,lat} w_i(lon, lat)} \tag{2}$$


The GMSL differences time series are plotted over the S3A period between S3A and Jason-3, S3A and SARAL/AltiKa, Jason-
3 and SARAL/AltiKa (Fig.1, a) and over the S3B period between S3B and S3A, S3B and Jason-3, S3B and SARAL/AltiKa,
Jason-3 and SARAL/AltiKa (Fig.1, b). They obviously indicate different trends and therefore relative GMSL drifts between
these different altimetry missions. The objective of the study is to accurately estimate these relative GMSL drifts and their
associated uncertainties.

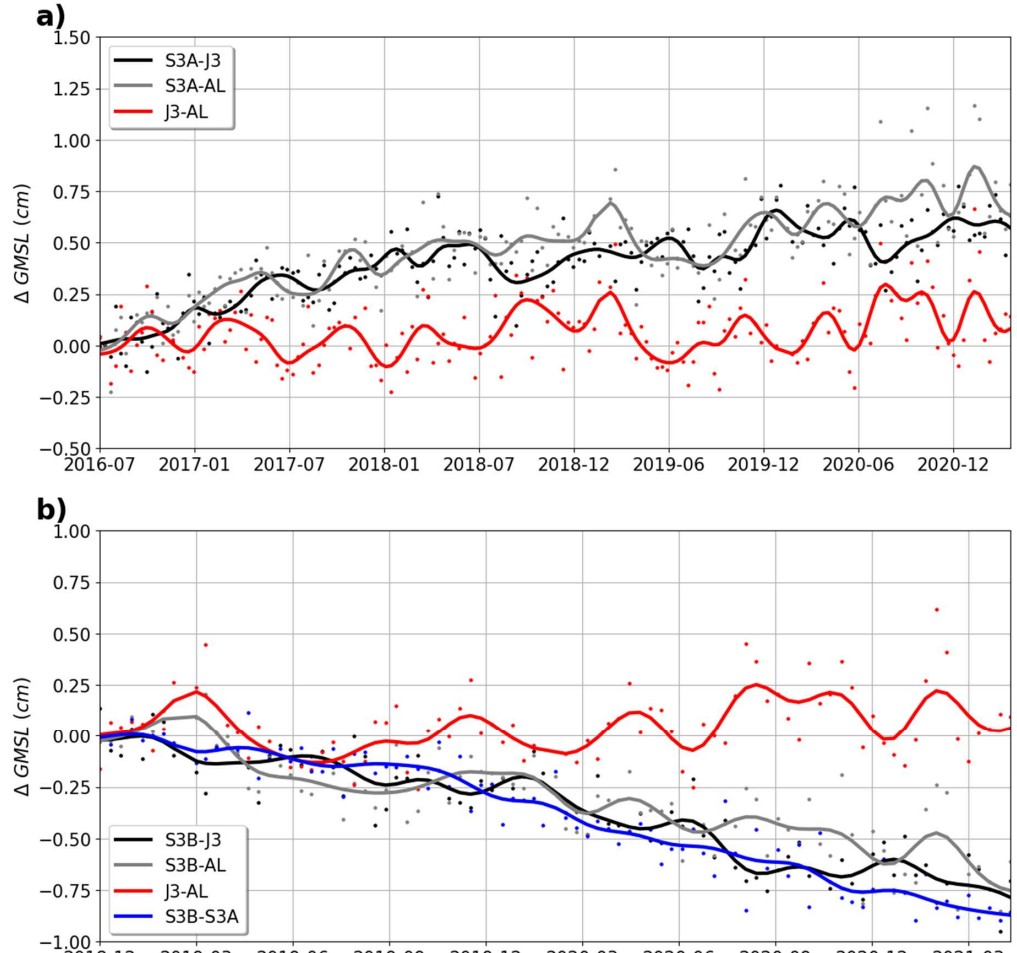


**Figure 1: Evolution of ΔGMSL: a) over the S3A period between S3A and Jason-3, S3A and SARAL/AltiKa, Jason-3 and
SARAL/AltiKa; b) over the S3B period between S3B and S3A, S3B and Jason-3, S3B and SARAL/AltiKa, Jason-3 and
SARAL/AltiKa. The dotted curves are the raw time series sampled at 10-days. The solid lines are time series filtered at 3 months
with a low pass filter. Each time series is artificially set to 0 at its origin.**

Other cross-calibration methods could be used in order to estimate the GMSL drift. Among them, the comparison of altimetry
and in-situ tide gauge (TG) measurements is often used to estimate the GMSL drift estimated from altimetry (Mitchum, 1998;
Valladeau et al., 2012; Watson et al., 2015, 2021). Although this method provides very relevant estimates of GMSL drifts for
long periods (> 10 years), it is not very suitable for shorter and more recent periods. On the one hand, the delay in the



availability of the global tide gauge network data (e.g. GLOSS/Clivar) is often more than 1 year, and does not allow
comparisons with the most recent altimetry data. On the other hand, the uncertainty associated with the calculation of the
GMSL drift with this method is large for short periods. It is of the order of 1.5 mm yr$^{-1}$ over 3 years and 1 mm yr$^{-1}$ over 5 years
(Ablain, 2018; Watson et al., 2021) within a confidence level of 90% (1.65-σ). This method therefore fails to detect a drift of
a few tenths of a mm yr$^{-1}$ over the periods of interest in our study. We show in the following sections that the chosen method,
i.e., direct altimetry mission comparison, provides more accuracy than the TG-altimetry method. However, comparison to tide
gauges allows us to obtain an estimate of the GMSL drift with independent measurements. For information purposes, we have
therefore provided these values in the "results" sections.

## 3.2. GMSL drift estimate and uncertainty

In order to estimate the relative GMSL drifts between altimetry missions compared two by two, a rigorous approach is
proposed. The first step is to compute the variance-covariance matrix ($\Sigma$) of the $\Delta GMSL$ time series errors, which is detailed
in depth later in this section. The second step consists in fitting the trend from a linear regression model ($y = X\beta + \epsilon$) applying
an Ordinary Least Square (OLS) approach, where the estimator of $\beta$ with the OLS, noted $\hat{\beta}$, is:

$$\hat{\beta} \sim (X^t X)^{-1} X^t y \qquad (3)$$


and where the distribution of the estimator $\hat{\beta}$ takes into account $\Sigma$ and follows a normal law:

$$\hat{\beta} = \mathcal{N}(\beta, (X^t X)^{-1} X^t (X^t X)^{-1}) \qquad (4)$$


This mathematical formalism was fully described in Ablain et al. (2019) to estimate the uncertainties of the GMSL trend and
acceleration. It is applied in this study to derive the $\Delta GMSL$ trend and its realistic uncertainty.
The calculation of $\Sigma$ is performed from the description of the errors of the GMSL differences between 2 altimeter missions
($\Delta GMSL$). We have applied the same approach as in Ablain et al. (2019), where the $\Delta GMSL$ error budget is composed of
different type of errors: a) drifts in $\Delta GMSL$ characterized by a trend uncertainty (±δ); and (b) time-correlated errors
characterized by their standard deviation (σ) and by the correlation timescale (λ). The values of the error budget are deduced
from those of the GMSL error budget over the Jason-3 period, and are presented in Tab.1.











| Source of errors | Error category | Uncertainty[1] level on 10-day cycles | | Additional information for the error budget on $\Delta GMSL$ |
|---|---|---|---|---|
| | | GMSL[2] | $\Delta$GMSL | |
| High frequency errors: altimeter noise, geophysical corrections, orbits | Correlated errors ($\lambda$ = 2 months) | $\sigma \in [1, 1.4]$ mm | $\sigma \in [0.6, 0.8]$ mm | Estimated directly from noise on GMSL time series, depends on altimetry missions (see section 3.2) |
| Medium frequency errors: geophysical corrections, orbits | Correlated errors ($\lambda$ = 1 year) | $\sigma \in [1, 1.2]$ mm | $\sigma \in [1, 1.2]$ mm | |
| Low frequency errors: wet tropospheric correction (WTC) | Correlated errors ($\lambda$ = 5 years) | $\sigma$ = 1.1 mm | $\sigma$ = 0 | Model WTC errors are cancelled out |
| Low frequency errors due to gravity fields in orbit calculation | Correlated errors ($\lambda$ = 10 years) | $\sigma$ = 0.5 mm | $\sigma = \sqrt{2} * 0.5$ mm | Orbit errors are assumed uncorrelated |
| Long-term drift errors due to ITRF in orbit calculation | Drift error | $\delta$ = 0.1 mm yr$^{-1}$ | $\delta = \sqrt{2} * 0.1$ mm yr$^{-1}$ | |
| Long-term drift errors: orbit GIA | Drift error | $\delta$ = 0.05 mm yr$^{-1}$ | $\delta$ = 0 | GIA errors is cancelled out |

[1] *All uncertainties reported are Gaussians, and they are given at the 1-$\sigma$ level*
[2] *The GMSL error budget is from the study by Ablain et al. (2019).*

**Table 1: Error budget on $\Delta$GMSL between 2 altimeter missions ($\Delta$GMSL) derived from the GMSL error budget from Ablain et al. (2019).**

Except for altimeter or radiometer induced drifts, which are totally independent between missions, or orbit induced drifts which
can also be totally independent, the drifts that may occur in the GMSL record are atmospheric corrections or tidal corrections
that are common to all altimetry missions and are therefore mostly cancelled-out in the $\Delta GMSL$ timeseries. For instance, the
glacial isostatic adjustment correction is derived from the same model (Spada, 2017) for all the missions, and does not depend
on the altimeter mission characteristics; the error related to the global mean of this correction is then cancelled out by
comparing GMSL time series. On the other hand, the drift of the realization of the International Terrestrial Reference Frame
(ITRF) in which the altimeter orbits are determined provided by Couhert et al. (2015) ($\delta$ = 0.1 mm yr$^{-1}$), is assumed to be
uncorrelated between 2 missions that are not on the same orbit (for example S3A and Jason-3). In this case, the uncertainty
level of $\Delta GMSL$ corresponds to the sum of the variance of the error orbit uncertainty in GMSL ($\delta = 0.1 * \sqrt{2}$ mm yr$^-$).
In the GMSL error budget, the residual time-correlated errors are separated in two parts: 1) errors with short correlation
timescales, i.e. lower than 1 year, 2) and errors with longer correlation timescales between 5 and 10 years. For the first part,
errors in GMSL are mainly due to the geophysical corrections (ocean tides, atmospheric corrections), to the altimeter
corrections (sea-state bias correction, altimeter ionospheric corrections) and the orbital calculation. As the altimeter sea-level
is calculated homogeneously for all the altimeter missions in this study (e.g. same ocean tide model, same wet tropospheric
correction from model), a significant part of these errors is cancelled out in GMSL differences. The remaining uncorrelated



errors come from orbital solutions whose errors are independent between altimetry missions at these short time scales. Residual
errors in some orbit repeatability-dependent corrections (e.g. aliasing of the ocean tide correction as 58.77-day signal
(Zawadzki and Ablain, 2016)) may also still be present in the $\Delta GMSL$ timeseries. Another error contribution is coming from
the oceanic variability (e.g. mesoscale) differently observed at short time scales by each altimer mission due to their different
orbits characteristics (Dufau et al., 2016). This error description allows us to consider all high frequency content of the GMSL
time series lower than 1 year as an error signal. The error signal variance is empirically estimated by measuring the variance
of the GMSL time series for signals lower than 1 year. Following the approach proposed in Ablain et al. (2019), we split the
variance estimate for high frequency signal (lower than 2 months) and medium frequency signal (between 2 months and 1
year) in order to better represent the frequency content of the error signal, which is higher at high frequencies, in particular
because of the mesoscale signal (< 2 months) observed differently by the altimetry missions. However, the choice of the 2-
month cut-off period to separate the high and medium frequencies is somewhat arbitrary. In section 4.2, we have evaluated the
sensitivity of varying this period from 1 month to 6 months, in order to assess the impact on the drift uncertainty estimate,
especially over short periods.
For the second part of residual time-correlated errors, between 5 and 10 years, errors in GMSL time series are due, on the one
hand, to instabilities in the wet tropospheric correction (Legeais et al., 2018) derived from on-board microwave radiometers,
and on the other hand, to the gravity field errors in orbit calculation (Couhert et al., 2015). In $\Delta GMSL$ timeseries, the wet
tropospheric correction errors are cancelled out since we have applied the same correction for all the altimeter missions derived
from the ECMWF model (see section 2). For the gravity field errors in orbit calculation ($\sigma = 0.5$ mm), they are assumed to
be uncorrelated between 2 altimeter missions that are not on the same orbit (for example S3A and Jason-3). In this case, the
uncertainty level $\Delta GMSL$ time series is the sum of the variance of the GMSL error budget uncertainty ($\sigma = 0.5 * \sqrt{2}$ mm).
The error variance-covariance matrix ($\Sigma_{\Delta GMSL}$) is then inferred from the $\Delta GMSL$ error budget for each couple of altimeter
missions (e.g; S3A and Jason-3) over the S3A and S3B periods. In short, the elementary variance-covariance matrices ($\Sigma_{Error_i}$
) corresponding to each error described in the $\Delta GMSL$ error budget are first calculated independently of each other. Each matrix
is calculated from a large number of random draws (> 1000) of simulated error signals whose correlation is modelled. Their
shape depends on the type of errors prescribed (e.g. time-correlated errors, long-term drifts). Assuming errors are independent,
$\Sigma_{\Delta GMSL}$ is given by the sum of all $\Sigma_{Error_i}$.

### 3.3. Extension of the method at local scales

It is quite straightforward to extend the approach proposed at global scale to derive the $\Delta GMSL$ drifts and uncertainties, to
local scales. The first step consists in calculating the local Mean Sea Level differences (noted $\Delta MSL$ hereafter) by averaging
the 3°x1° lon/lat SLA grid (see section 3.1) at different spatial scales. For this study, we arbitrarily chose different cell sizes
in order to calculate the local MSL drifts and its associated uncertainties at different local spatial scales: 3°x3° (~240 km),
9°x9° (~700 km) and 30°x30° (~2400 km). The second step consists in calculating the local $\Delta MSL$ error budget from the local





MSL error budget from Prandi et al. (2021), following the same approach as for the $\Delta GMSL$ error budget (section 3.2). The
updated values of the $\Delta MSL$ error budget are presented in Tab.2. In similar fashion to the $\Delta GMSL$ error budget, the GIA
induced drift and low frequency wet tropospheric correction (using model WTC) errors are cancelled out. Prandi et al. 2021
evaluate the long term orbit errors that affect regional MSL at $\delta = 0.33$ mm yr$^{-1}$. Assuming that those errors are independent
between 2 altimeter missions on different orbits, the uncertainty level of the local $MSL$ time series is the sum of the variance
of the local MSL error budget uncertainty: $\delta = 0.33 * \sqrt{2}$ mm yr$^{-1}$. For the evaluation of the uncertainty level of short time
scale errors, the variance of the error signal is evaluated from the high frequency content lower than 1 year of local $\Delta MSL$ time
series, and the variance estimate is splitted between a high frequency signal (lower than 2 months) and a medium frequency
signal (between 2 months and 1 year) to obtain a better frequency representation of the signal. We obtain a location-dependent
error signal for high and medium frequencies (see supplementary material). The standard deviation of the high frequency
signal below 2 months ranges between 13.3 and 30.7 mm, highlighting the signature of the mesoscale in the large ocean
currents. For medium frequencies (between 2 months and 1 year), the variations are weaker: between 6.9 and 17.7 mm. They
are also more homogeneous, and with a low signature of large ocean currents.

| Source of errors | Error category | Uncertainty[1] level on 10-day cycles (1- $\sigma$) | | Additional information for the error budget on local $\Delta$MSL |
| --- | --- | --- | --- | --- |
| | | Local MSL[2] | Local $\Delta$MSL | |
| High frequency errors: altimeter noise, geophysical corrections, orbits | Correlated errors ($\lambda$ = 2 months) | Location dependent. | Location dependent. $\sigma \in [13.3, 30.7]^{(3)}$ mm | Estimated directly from noise on local MSL difference time series, depends on altimetry missions. (see section 3.3) |
| Medium frequency errors: geophysical corrections, orbits | Correlated errors ($\lambda$ = 1 year) | Location dependent. | Location dependent. $\sigma \in [6.9, 17.7]^{(3)}$ mm | |
| Low frequency errors: wet tropospheric correction (WTC) | Correlated errors ($\lambda$ = 5 years) | Location dependent. | $\sigma = 0$ | Model WTC errors are cancelled out |
| Long term drift errors : orbits | Drift error | $\delta = 0.33$ mm yr$^{-1}$ | $\delta = \sqrt{2} * 0.33$ mm yr$^{-1}$ | Orbit errors are assumed uncorrelated |
| Long-term drift errors: GIA | Drift error | Location dependent | $\delta = 0$ | GIA errors is cancelled out |

$^{(1)}$ *All uncertainties reported are Gaussians, and they are given at the 1-$\sigma$ level*

$^{(2)}$ *The local MSL error budget is from the study by Prandi et al. (2021)*

$^{(3)}$ *Values provided for 3°x3° box sizes within a 16th-percentile and 84th-percentile interval.*

**Table 2: Error budget on MSL differences at local scale between 2 altimeter missions (ΔMSL) derived from the MSL error budget**
**at local scale from Prandi et al. (2021).**



**4. Results**

**4.1. S3A GMSL drift detection**

The $\Delta GMSL$ trend and uncertainty is computed using the method provided in section 3.2 between S3A and Jason-3, S3A and SARAL/AltiKa, Jason-3 and SARAL/AltiKa, on a common period between July 2016 and March 2021. Fig.2 shows the $\Delta GMSL$ trends, and trend uncertainties at 68% C.L. (1-σ, in black) and 90% C.L. (1.65-σ in grey), for each of those pairs: 1.01 ± 0.31 mm yr⁻¹ between S3A and Jason-3; 1.28 ± 0.37 mm yr⁻¹ between S3A and SARAL/AltiKa; and 0.29 ± 0.41 mm yr⁻¹ between Jason-3 and SARAL/AltiKa. Calculating the ratio between the $\Delta GMSL$ trend and the associated uncertainty, we can indicate the confidence level in which the relative $\Delta GMSL$ trend is measured. Between S3A and Jason-3, as well as between Jason-3 and SARAL/AltiKa, the confidence level at which a trend is detected is 99.9% (corresponding to 3.4-σ). However, a trend between Jason-3 and SARAL/AltiKa is only measured with a low 57.0 % confidence level, and furthermore the estimated $\Delta GMSL$ trend value is small (0.29 mm yr⁻¹), compared to the S3A relative $\Delta GMSL$ trend with both Jason-3 and SARAL/AltiKa.

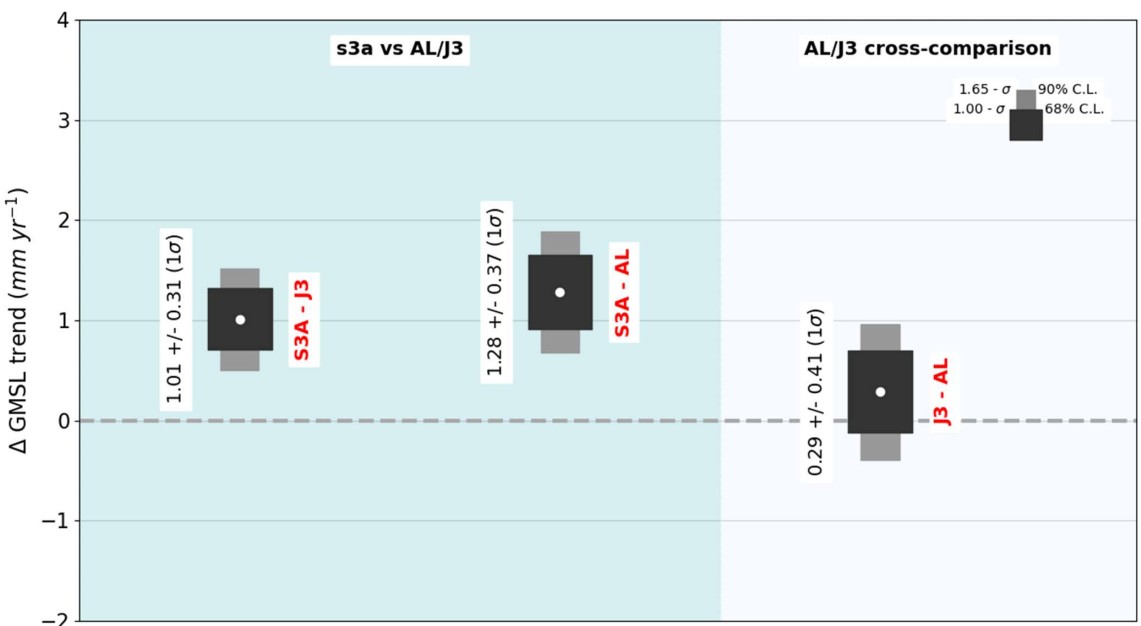

**Figure 2: △GMSL trend differences between S3A and Jason-3, SARAL/AltiKa, over the july 2016 to march 2021 period. The black boxes show the △GMSL trend uncertainties at 68% C.L. and the grey boxes at 90% C.L.**

These results highlight a significant $\Delta GMSL$ trend between S3A and Jason-3 as well as with SARAL/AtiKa, whereas Jason-3 and SARAL/AltiKa are in agreement within the confidence level. This result likely suggests a drift in the S3A GMSL.





Moreover, this result is also confirmed by the comparison with Jason-2, albeit over a shorter period due to the shutdown of
Jason-2 in September 2017 (not presented in the paper). The $\Delta GMSL$ trend obtained between S3A and Jason-2 is $4.45 \pm 0.98$
mm yr$^{-1}$ over the March 2016 to September 2017 period. On the same period, the $\Delta GMSL$ trends obtained between S3A and
Jason-3 and SARAL/AltiKa are respectively $3.66 \pm 0.93$ mm yr$^{-1}$ and $2.83 \pm 1.16$ mm yr$^{-1}$. Although the trend uncertainties
are higher over this shorter period, the $\Delta GMSL$ trends are still significant. Those results also indicate that the S3A GMSL drift
may have been stronger during its first year of operations. It is confirmed by the analysis on the S3B period (December 2018
to March 2021) in next section 4.2 where Fig.3 exhibits lower $\Delta GMSL$ trends of $0.66 \pm 0.62$ mm yr$^{-1}$ between S3A and Jason-
3, and $1.38 \pm 0.90$ mm yr$^{-1}$ between S3A and SARAL/AltiKa.
The very likely drift in the S3A GMSL is also observed through independent comparisons with tide gauges, using the method
provided by Valladeau et al., 2012 and Ablain, 2018, and data from the GLOSS/CLIVAR tide gauge network. Over the July
2016 to December 2020 period, a significant relative GMSL drift of $1.18 \pm 0.63$ mm yr$^{-1}$ (1-σ) is also detected between S3A
and the GC tide gauges network.
All of these consistent analyses reveal that a drift in the S3A GMSL between 1.0 and 1.3 mm yr$^{-1}$ $\pm 0.3$ mm yr$^{-1}$ is most likely
detected. However, the S3A GMSL drift is much larger than the 0.3 mm yr$^{-1}$ GMSL PTR-induced drift anticipated by Poisson
et al., 2019. Thanks to the results of this study, carried out in the frame of the Sentinel-3 MPC (Mission Performance Centre)
project supported by ESA, further studies supported by CNES succeeded to explain the remaining part of the S3A GMSL drift
(about ~0.7-1.0 mm yr$^{-1}$, Aublanc et al. 2020). This drift is due to some inner features of the S3 SAR processing, not properly
considered. A correction, so-called 'range walk' correction (not detailed in this paper) was proposed by Aublanc (2020) that
will be implemented in the S3 altimeter ground processing chain in early 2022. It is also interesting to note that the 0.3-0.4
mm yr$^{-1}$ contribution of PTR-induced S3A-GMSL drift is not detectable with a sufficient confidence level over such a short
period. One would need a 5-year period to detect a drift of about 0.3 mm yr$^{-1}$ with a confidence level of about 60-70%.
**4.2. S3B GMSL drift detection**
In exactly the same fashion as for S3A, the $\Delta GMSL$ trends and associated uncertainties are computed between S3B and Jason-
3, S3B and SARAL/AltiKa, S3B and S3A, S3A and Jason-3, S3A and SARAL/AltiKa, and Jason-3 and SARAL/AltiKa, on a
common period between December 2018 and March 2021 i.e. 2 years and 4 months. Fig.3 represents the $\Delta GMSL$ trends, and
trend uncertainties at 68% C.L. (1 σ, in black) and 90% C.L. (1.65 σ, in grey), for each of those pairs.
We can first note that a strong and significant negative $\Delta GMSL$ trend is exhibited when S3B is compared to all three other
missions: $-3.44 \pm 0.61$ mm yr$^{-1}$ between S3B and Jason-3; $-2.76 \pm 0.77$ mm yr$^{-1}$ between S3B and SARAL/AltiKa; $-4.09 \pm$
$0.52$ mm yr$^{-1}$ between S3B and S3A. $\Delta GMSL$ trends are significant within a confidence level over 99.9%. In the meantime,
$\Delta GMSL$ trends without S3B are much smaller and more consistent over the S3B period: $0.66 \pm 0.61$ mm yr$^{-1}$ between S3A and
Jason-3; $1.38 \pm 0.90$ mm yr$^{-1}$ between S3A and SARAL/AltiKa; $0.64 \pm 0.91$ mm yr$^{-1}$ between Jason-3 and SARAL/AltiKa.
Therefore, these results allow us to state that the detection of a drift of the S3B GMSL is very likely with a minimum value of
$-2.22$ mm yr$^{-1}$ and a maximum value of $-4.05$ mm yr$^{-1}$ within a confidence level of 99%. Furthermore, these results are also



confirmed by tide gauge comparisons that indicate a significant drift of the S3A GMSL of -4.04 mm yr⁻¹ ± 1.45 mm yr⁻¹ (1-σ)
over the December 2018 to December 2020 period. This drift is quite surprising since the S3B altimeter mission is very similar
to S3A (same altimeter, same configuration). To date, this drift is under investigation by S3B altimetry experts, but remains
unexplained.

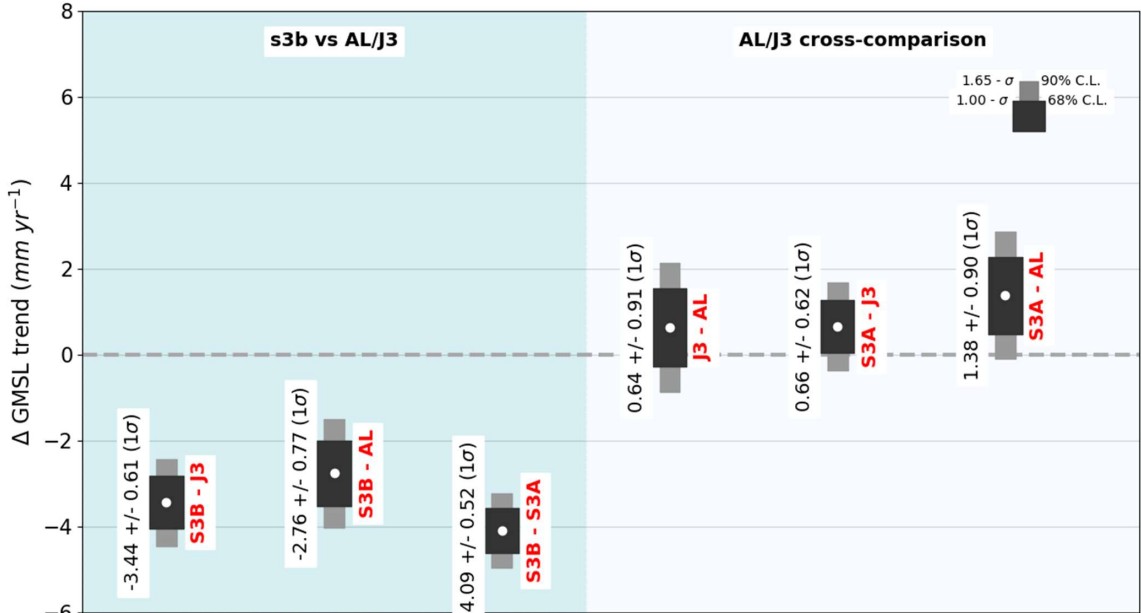


**Figure 3: △GMSL trend differences between S3B and Jason-3, S3A, SARAL/AltiKa, over the December 2018 to march 2021 period.**
**The black boxes show the △GMSL trend uncertainties at 68% C.L. and the grey boxes at 90% C.L.**


### 4.3. △GMSL trend uncertainty estimates versus the period length

In order to accurately estimate the ability of the proposed method to detect a significant relative drift between 2 missions, we
calculated the evolution of these uncertainties as a function of the period length. In Fig.4, the black plain line shows the
theoretical evolution of the $\Delta GMSL$ trend uncertainty between S3A and Jason-3 for period lengths from 1 year to 10 years,
using the error budget presented in Tab.1. The $\Delta GMSL$ trend uncertainty evolves from 1.75 mm yr⁻¹ for a 1-year period, quickly
decreases to 0.5 mm yr⁻¹ for a 3-year period before slowing down to reach 0.3 mm yr⁻¹ a for 5-year period, and finally converges
to 0.2 mm yr⁻¹ for a 10-year period. The knowledge of the statistical behaviour of the errors (Tab.1) is a difficult task,
performed under certain assumptions (see section 3.2). We have therefore tested the sensitivity of our uncertainty calculations.
Firstly, in Tab.1, we have assumed that the GMSL drift caused by the ITRF realization in orbit calculation is uncorrelated
between 2 altimetry missions. It is however very likely that this error is strongly correlated even if this information is not
quantified in the literature. We have therefore tested the impact of cancelling out this error assuming this time that it is fully
correlated between 2 measurements. The uncertainty level obtained is displayed with the black dotted line in Fig.4. For a 5-
year period, the uncertainty is reduced to 0.27 mm yr$^{-1}$ (instead of 0.3 mm yr$^{-1}$), and for a 10-year period, it is reduced to 0.13
mm yr$^{-1}$ (instead of 0.2 mm yr$^{-1}$). This result has no impact on our study since we have considered the most conservative
approach, i.e. the one which yields the highest uncertainties.




**Figure 4: Evolution of △GMSL trend uncertainties versus period length, from the S3A and Jason-3 comparison. The black solid**
**line is the △GMSL trend uncertainty derived from the △GMSL error budget (Tab.1). The back dotted lines is the △GMSL trend**
**uncertainty derived from the △GMSL error budget (Tab.1) with the orbit ITRF error contribution set to 0. The red envelope is the**
**dispersion of △GMSL trend uncertainty between the 5th-percentile and 95th-percentile (i.e. 1.65-sigma) by varying the cut-off**
**frequency of the high frequency errors from 0.5 to 6 months.**

We have also evaluated the sensitivity of the prescription of high and medium frequency errors lower than 1 year. As mentioned
in section 3.1, the choice of the 2-month cut-off period is based on the assumption that mesoscale signals are uncorrelated over
periods larger than 2 months, but it is somewhat arbitrary. Thus, we have varied the cut-off period for a range of periods from
0.5 months to 6 months. The red envelope shown in Fig.4 represents the dispersion of △GMSL trend uncertainty obtained
between the 5th-percentile and 95th-percentile (i.e. 1.65-sigma). While the variations of the uncertainties can be considered
negligible for time periods above 5 years (< 0.1 mm yr$^{-1}$ ), they are more important for shorter time periods where they reach
0.35 mm yr$^{-1}$ and 0.2mm yr$^{-1}$ for time periods of 2 and 3 years respectively. Given the sensitivity range of our method to





estimate uncertainties for periods of 4 years and 9 months (S3A) and 2 years and 4 months (S3B), the drifts observed on Fig.2
and Fig.3 are still significant. Our conclusions are thus unchanged. However, one should pay attention to it for studies over
very short periods of time (<3 years). In addition, it would be relevant to develop other approaches (e.g. based on Fourier
analysis) to evaluate the high frequency spectral content of $\Delta GMSL$ time series.
**4.4. S3A and S3B local sea level drift detection**
Applying the method described in section 3.3, we evaluated the local $\Delta MSL$ trends and their uncertainties for S3A and S3B,
compared to Jason-3 and SARAL/AltiKa, for different spatial scales : 3°x3° cells of ~240km length, 9°x9° cells of ~700km
length, and 30°x30° cells of ~2400km length. Local △MSL trends are represented in Fig.5 (a) for S3A and Jason-3 differences
on 9°x9° cells (~240km regional scale) after removing the global mean trend (i.e. 1.13 mm yr$^{-1}$). Local △MSL trends are
ranging from -2 and +2 mm yr$^{-1}$ with higher values in main large ocean currents (e.g. Kurushio). In contrast, we do not
distinguish large geographically correlated spatial structures. They might have indicated systematic local biases in the MSL
trends on either of the 2 missions.
The confidence level of the measured local △MSL trends can be obtained by dividing the absolute value of the local △MSL
trend by the associated trend uncertainty for each cell. When this ratio is less than 1, the local △MSL trend is less than the
uncertainty associated with 1-σ and is therefore estimated with a confidence level less than 68%, i.e., very unlikely. When the
ratio is between 1 and 2, the local △MSL trend is estimated with a confidence level between 68% and 95%, i.e., likely. When
the ratio is greater than 2, the local trend △MSL is estimated with a confidence level greater than 95%, i.e., very likely. The
ratio is represented in Fig.5 (b) for S3A and Jason-3 differences. We observe that none of the local △MSL trend are significant
since they are measured with less than 68% confidence level. We have performed the same analyses with different size boxes
until 30x30° degrees (i.e. 2400 km), and we do not detect any significant local △MSL trend. We also obtain similar results
calculating the local △MSL trends between S3B and Jason-3, where we cannot detect any significant trends (see figure in
auxiliary materials).

**a)**

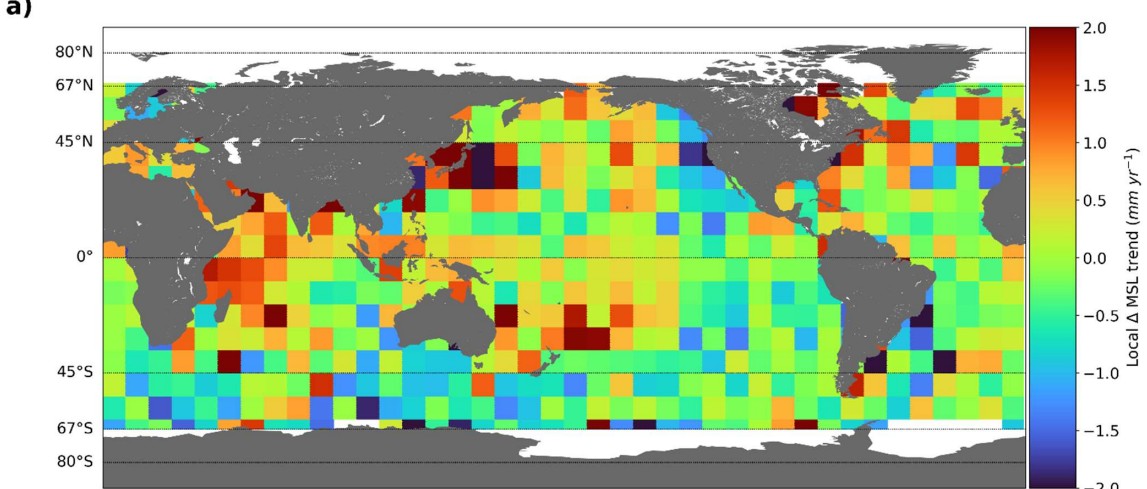

**b)**

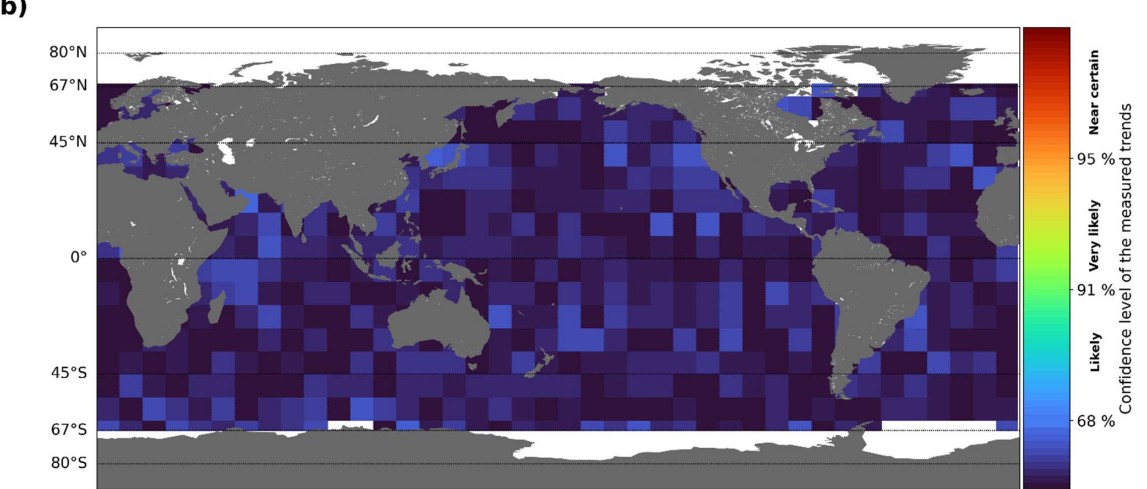

**Figure 5: a) Local △MSL trends between S3A and Jason-3 after removing the global mean trend (i.e. 1.13 mm yr⁻¹ in 9°x9° resolution. b) Confidence level of the measured Local △MSL trends computed from local △MSL trends divided by local uncertainties between S3A and Jason-3 in 9°x9° resolution.**

The fact that no significant local trend △MSL is detected between S3A and Jason-3, and between S3B and Jason-3, does not demonstrate the absence of local MSL drift on these altimetry missions. However, this indicates that the level of uncertainty associated with the method implemented is too high to allow the detection of significant trends between these missions. We represent on Fig.6 the evolution of the local △MSL trend uncertainties versus the period length, for the three spatial scales





considered, based on the S3A and Jason-3 comparison. For a 3-years period, a local △MSL trend  over 2.5 mm yr$^{-1}$ can be
detected for the larger 2400 km regional scale (30°x30°), and respectively 5 mm yr$^{-1}$ and 10 mm yr$^{-1}$ for the 700 km (9°x9°)
and 240 km (3°x3°) local scales. For a 5-year period, which is a typical period for which two altimetry missions are in orbit
simultaneously, a local △MSL trend over 1.5 mm yr$^{-1}$ can be detected for the larger 2400 km regional scale (30°x30°), and
respectively 2.5 mm yr$^{-1}$ and 5 mm yr$^{-1}$ for the 700 km (9°x9°) and 240 km (3°x3°) regional scales. These figures correspond
to a global average but may change locally depending on the high-frequency content of the MSL differences provided in Tab.2.
The envelopes displayed in Fig.6 represent the 16th and 84th percentile, corresponding to 1-σ of the spatial distribution of
△MSL trend uncertainties across the oceans. These envelopes show that the uncertainties vary locally a lot (e.g. between 1
and 3 mm yr$^{-1}$ for a 5-year period and 2400 km box lengths). These spatial variations are mainly due to the mesoscale signal,
which is not observed in the same way by altimetry missions (see supplementary material). The lowest level of local uncertainty
obtained is 0.75 mm yr$^{-1}$ with spatial variations between 0.6 and 1.1 mm yr$^{-1}$, considering boxes of 2400 km over a 10-year
period.

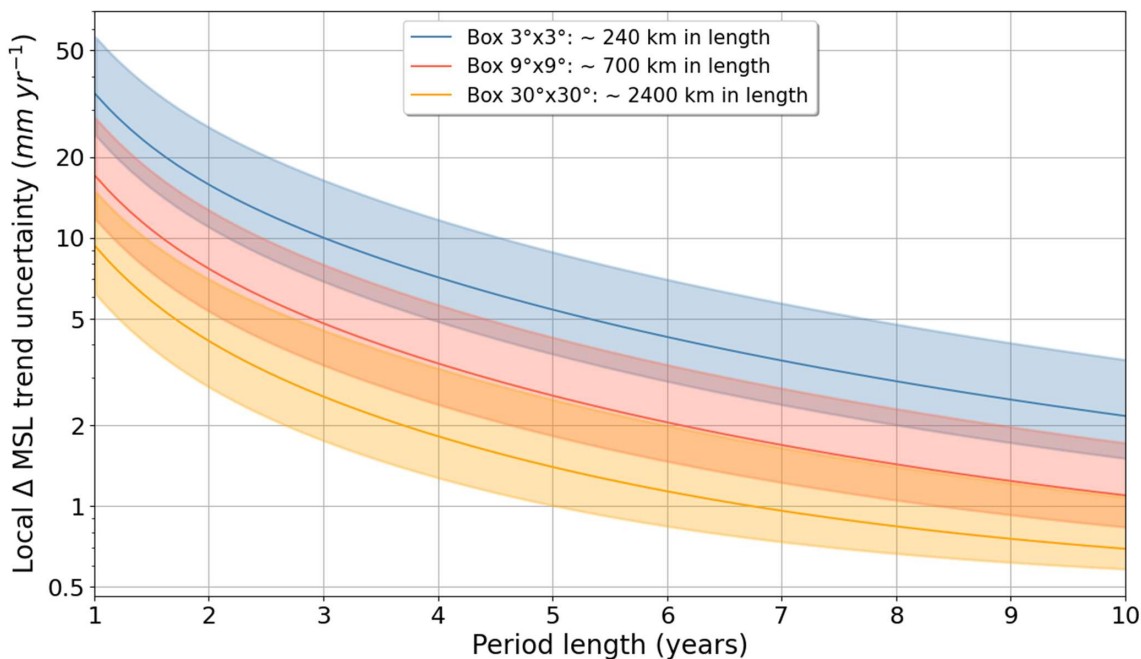


**Figure 6: Evolution of the S3A - Jason-3 local △MSL trend uncertainties as a function of period length for different box sizes. The**
**solid line is the global median of local △MSL trend uncertainties. The envelope represents the spatial distribution of uncertainties**
**at between the 16th and 84th percentile (i.e. 1-sigma) values. Y-axis scale is logarithmic.**





## 5. Conclusions

In this study, we have very likely detected a drift on the Copernicus S3A and S3B GMSL by implementing a method based on cross-comparison to Jason-3 and SARAL/AltiKa altimetry mission. We have also shown that no spatial variation of the global GMSL drift is detectable for either S3A or S3B, within the uncertainty level of the proposed method. For S3A, the detected relative GMSL drift is 1.0 mm yr$^{-1}$ with Jason-3 and 1.3 mm yr$^{-1}$ with SARAL/AltiKa, with more than 99% confidence level, over the July 2016 to March 2021 period. This relative drift is also observed with Jason-2 over a shorter period, as well as when compared to tide gauges. The S3A GMSL drift appears also stronger over the first year of operations: between 2.5 and 4 mm yr$^{-1}$ with a confidence level higher than 68%. Thanks to a close cooperation with altimeter experts (in the frame of the S3 MPC project supported by ESA), the origin of the drift is now mainly explained by both a drift on the S3A altimeter PTR parameters, responsible for about 0.3-0.4 mm yr$^{-1}$ (Poisson et al, 2018), and a drift due to wrong hypotheses used in the SAR processing (Aublanc, 2020). A correction proposed by Aublanc (2020) (so-called 'range walk' correction) will be implemented in the S3 altimeter ground processing chain in early 2022. For S3B, the detected relative GMSL drift is -2.8 mm yr$^{-1}$ with SARAL/AltiKa and -3.4 mm yr$^{-1}$ with Jason-3 with 99% confidence level, over the December 2018 to march 2021 period. The origin of the drift is still under investigation by the altimeter experts.

By detecting GMSL drifts on S3A and S3B, we have shown the ability of the implemented method to detect relative GMSL drifts for any period lengths. The typical order of magnitude of relative GMSL drifts that can be detected are the following: 0.5 mm yr$^{-1}$ for a 3-year period, 0.3 mm yr$^{-1}$ for a 5-year period, and 0.2 mm yr$^{-1}$ for a 10-year period. At local scales, relative MSL drift over 2.5 mm yr$^{-1}$ can be detected over a 3-year period, and up to 1.5 mm yr$^{-1}$ for a 5-year period for the larger local scales studied (2400 km).

Finally, the proposed cross-calibration method allows for the detection of sea-level drifts close to the GCOS requirements on sea-level stability (GCOS, 2011), which are 0.3 mm yr$^{-1}$ at the global scale and 1.0 mm yr$^{-1}$ at local scales over a minimum 10-year period. Our method is also significantly more accurate than the GMSL drift detected by comparison with tide gauges (Ablain, 2018; Watson et al., 2021): 0.8 mm yr$^{-1}$ over a 5-year period and 0.5 mm yr$^{-1}$ over 10-year period. However our proposed approach only detects uncorrelated drifts between missions (e.g. altimeter drift), and not the correlated drifts that might be present in orbit solutions or geophysical correction. Therefore, other approaches based on comparison with independent measurements such as global tide gauges network, are required to estimate sea-level drifts of the whole altimeter system. In addition, the comparison between two altimetry missions can be performed over a common period of often less than 8 years, while the comparison between altimeters and tide gauges can be performed over the entire life of an altimetry mission, since the launch of TOPEX/Poseidon in 1992.

Recently, Meyssignac (2019) has identified more stringent sea-level stability requirements for climate change studies of 0.1 mm yr-1 at global scale and 0.5 mm yr-1 at local scales. They cannot be met with our approach, even considering periods of 10 years, or more. We have shown that a better knowledge of the correlation of the orbit error between 2 altimeter missions should be investigated in more detail in future studies. Assuming this error is uncorrelated, we are approaching the GMSL



stability requirement of 0.1 mm yr$^{-1}$ over a 10-year period. Other approaches should also be considered to improve altimeter
sea-level drift detection. Ablain et al. (2021) proposed to perform two tandem phases between Jason-3 and S6-MF altimeter
missions. This particular configuration where the 2 satellites follow each other at less than a minute interval, allows for the
evaluation of the sea level drifts with an uncertainty of 0.1 mm yr$^{-1}$ at global scale, and 0.4 mm yr$^{-1}$ at the local scales, over a
3-years period only. However, this new approach, not yet implemented, is applicable only for satellites located on the same
orbit. For other satellite configurations, it would also be relevant to analyse cross-comparison methods based on measurement
selection at crossovers with a fairly restrictive time difference. This could possibly reduce the effect of oceanic variability in
sea level differences, and improve the detection of drifts, especially at local scales.
**Acknowledgement**
This study has been founded in the frame of the Sentinel-3 Mission Performance Centre (MPC) project supported by
ESA/ESRIN. We first acknowledge Sylvie Labroue (CLS) and Benoit Meyssignac (LEGOS) who provided support to this
study at its beginning. We also acknowledge Matthias Raynal from CNES for providing some input data and advice.





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





**Auxiliary materials**

**a)**

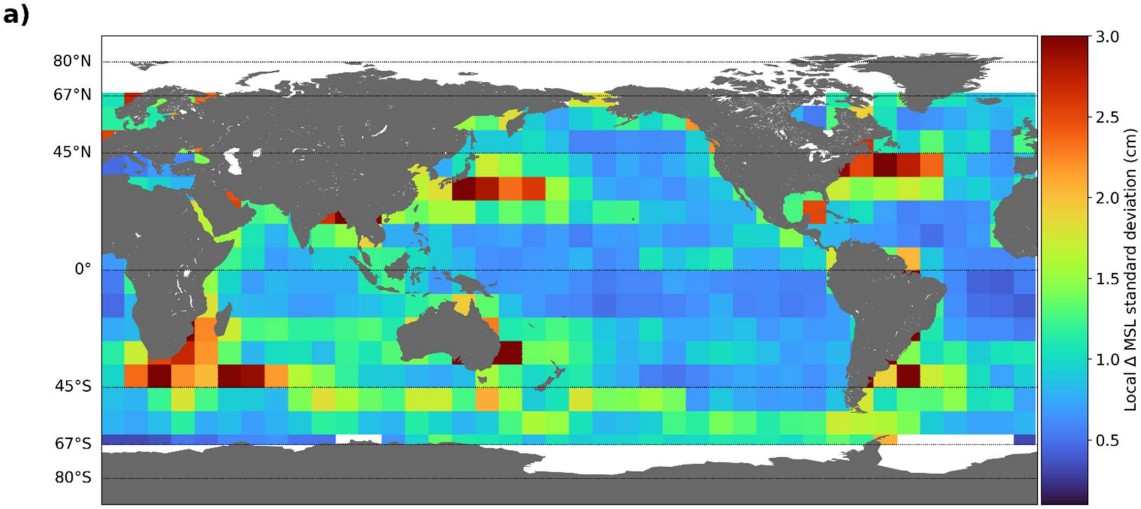

**b)**

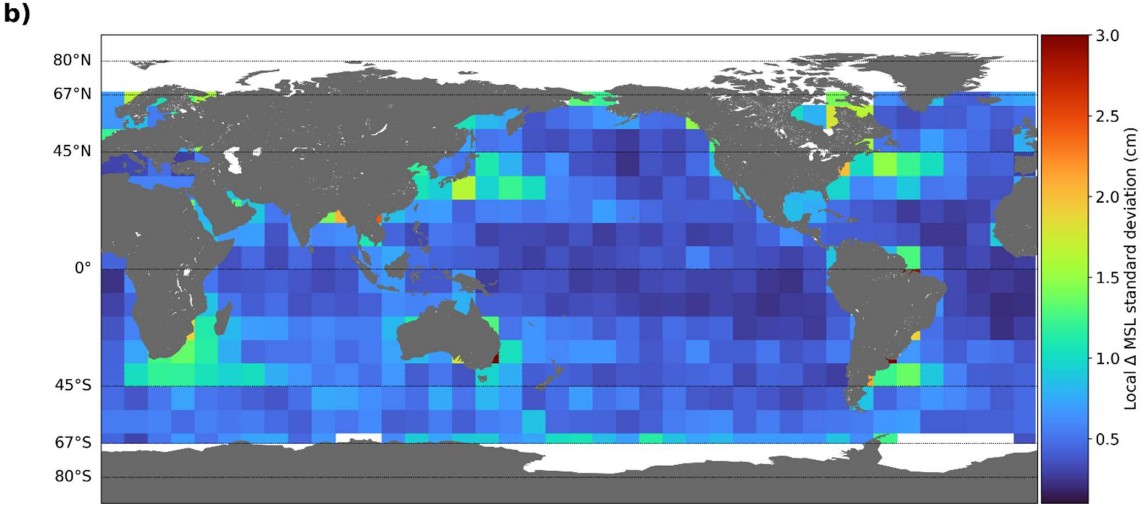


**Figure 7: a) Local △MSL high-frequency uncertainties (<2 months) between S3A and Jason-3 in 9°x9° resolution. b) Local △MSL**
**medium-frequency uncertainties (between 2 months and 1 year) between S3A and Jason-3 in 9°x9° resolution.**





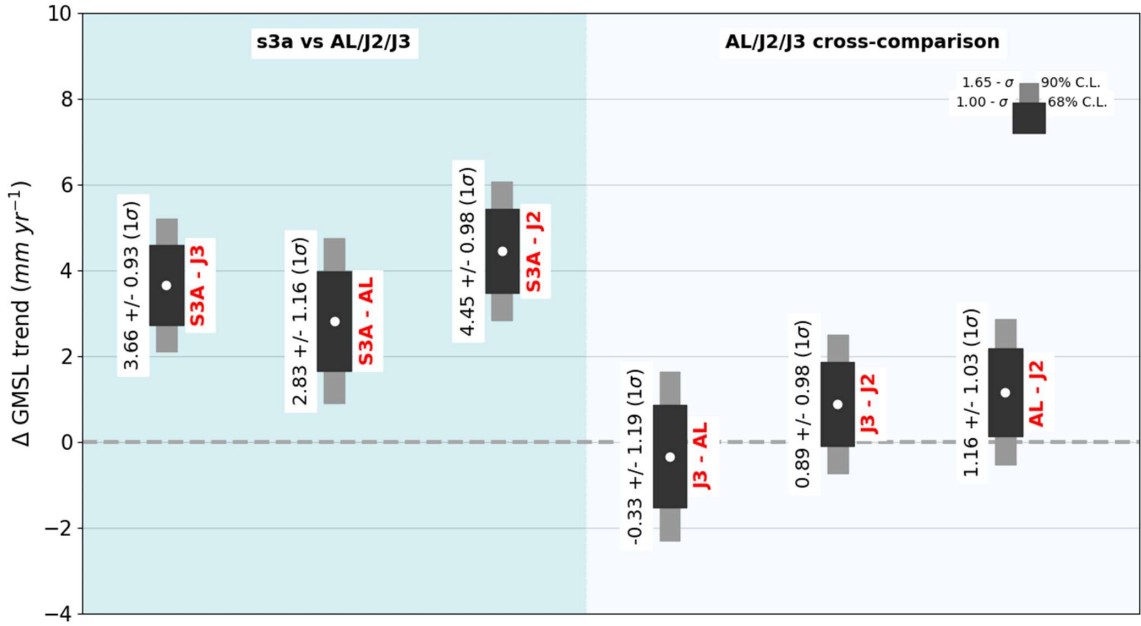

**Figure 8: △GMSL trend differences between S3A, Jason-2, Jason-3, and SARAL/AltiKa, over the March 2016 to September 2017 period. The black boxes show the △GMSL trend uncertainties at 68% C.L. and the grey boxes at 90% C.L.**

**a)**

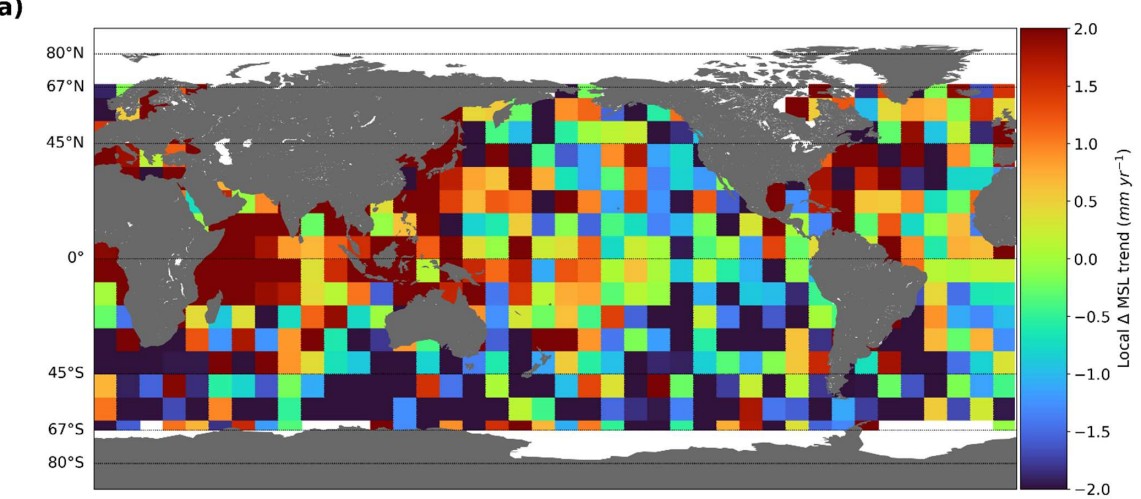

**b)**

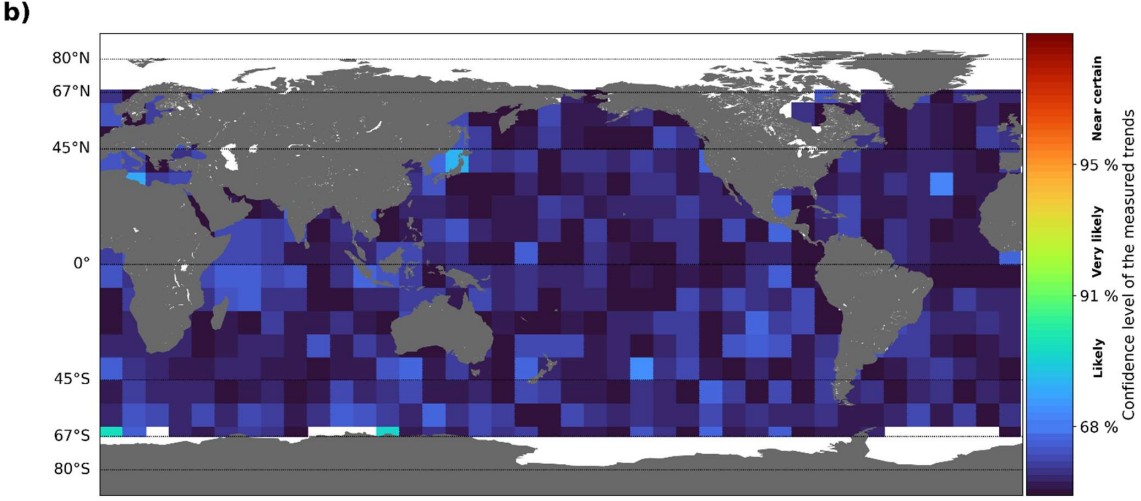

**Figure 9: a) Local △MSL trends between S3B and Jason-3 after removing (the global mean trend (i.e. -3.01 mm yr-1) removed from**
**the grid) in 9°x9° resolution. b) Confidence level of the measured Local △MSL trends Local drift probability computed from local**
**△MSL trends divided by local uncertainties between S3B and Jason-3 in 9°x9° resolution.**