# Peer review of "On the uncertainty associated with detecting global and local mean sea level drifts on Sentinel-3A and Sentinel-3B altimetry missions"

_Ocean Science, 2021_

## Author Comment (AC3)

**Response to first reviewer (Graham Quartly)**

Response: Thank you for your review, we have complied with most of the changes you've requested, except for reassessment of the S3B drift and providing an uncertainty assessment on crossover approach. They will both be evaluated in further studies and will therefore be the object of other future publications.

Kind regards,
Rémi Jugier

Sea level rise, and especially it's global average (GMSL) has been seen as one of the totemic measures of anthropogenic climate change, with accurate observations of the trend key to distinguishing between different scenarios. Satellite altimeters are the only means to provide a fully global picture, with similar trends noted from most missions (after much detailed analysis to understand instrumental effects!) The switch to delay/Doppler (or "SAR altimetry") for recent missions has raised questions about the consistency of trends from different satellites.

This paper provides some new ideas on how to assess the relative trends of missions, although it cannot identify which is the most accurate in an absolute sense. The method advocated is to interpolate each mission's data on a lat-lon-time grid, and then apply suitable area-based weighting to find the differences in trends. The main results are not new i.e. that S3A and S3B differ and show different trends to Jason-3. The new part is the discussion of methodology and the uncertainties.

It is unfortunate timing that this paper was submitted a few weeks before Dinardo revealed the cause of S3B's anomalous trend (incorrect application of USO correction). As a simple daily correction has been provided to undo the USO error, it would seem sensible to implement this and test whether there is now negligible drift between S3A and S3B. It could help if you mention the S3MPC overview paper (Quartly et al., 2020) noting the telemetered USO correction (Fig. 4b) and the other independent means used to monitor drift for S3A and S3B.

In the revised version of the paper, for S3B, we now explain the main cause of the drift (incorrect application of the USO correction), and link to Quartly et al. 2020. However, we

do not reassess the drift with the proper USO correction. As it was this study was the one (or one of) that initially managed to assess this drift, we think it is better to leave it as it is for proper tracking of the scientific process, and the assessment of the S3B drift with the correct application of the USO correction will be the object of future studies.

An alternative way to assess trend differences between just two missions e.g S3A and Jason-3) is to consider all dual crossovers < 10 days apart (which the authors briefly mention at the very end of the paper). I would like to see more discussion of the pros and cons of each method, along with a comparison of the uncertainties in each. It is not immediately clear that this new method is better. Indeed, we can consider crossovers < 10 days, or even < shorter periods. This approach is also very relevant : it allows for improving the spatial correlation between the measurements of the 2 missions but at the cost of fewer measurements being averaged. At this stage we have not yet evaluated the uncertainty of this approach to detect sea-level relative drifts between two altimeter missions. We plan to do this work on future studies with a similar statistical method to the one described in this paper. We would like to update this paper with a more general study that compares different cross-calibration methods together.

The discussion of how uncertainty varies with duration of the common period is very useful. In order to achieve accuracy in accord with GCOS requirements should the space agencies aim at 15-year missions rather than a series of satellites with 5-7 year lifetimes?

For Cal/ValL activities like the sea-level drift assessment presented in this paper, we need to have missions that are concurrently flying on a stable orbit track for the longer period possible. So, having short lifetime missions limits our ability to detect drifts, which can be partially offset if more missions are flying concurrently.  With this point of view, it would have a great interest to get space missions with a longer lifetime. We did not add this recommendation in the paper because we do not believe it is one of these objectives.

For the case when there is no consistent long-term error, but only errors with time scales of <2 months, I would expect the uncertainty in trend to vary with (Duration)^(-3/2). No mathematical form is provided, so is this the right scaling? Clearly with extra ITRF and GIA trend uncertainty, the value will tend asymptotically to ~0.2 mm/yr. Yes, the trend uncertainty indeed evolves as alpha * (Duration_adim)^(-3/2) for uncorrelated errors (white noise), and it is also true for correlated errors if Duration >> correlation period. However the 'alpha' value is higher for correlated errors than for uncorrelated errors due to the loss of degrees of freedom (see figure below). It is however interesting, from a purely theoretical standpoint, to note that for durations <= correlation period, as can be seen for 180 days period on the figure, the error stops rising at some point. This is

explained because, as the errors are completely correlated for all values, they are affected by any error in the same direction and it therefore does not affect the trend. We do not include this mathematical formula for the asymptotic behaviour because the period that mostly concerns us is the [0,6] years as missions rarely have overlapping periods for longer than this.

[Figure]

Trend uncertainty for an error budget containing only uncorrelated (0 days: black) or correlated (60 days: blue, 180 days: red) noise.

For a good part of this period SARAL/AltiKa was in a drifting orbit: is that why data are gridded at 2-monthly intervals rather than monthly? More information on the choice of processing options would be appreciated.

There is maybe a misunderstanding:  for all comparisons, we explain in section 3.1 that data is always gridded at 10 days intervals, not 1 or 2 months. Furthermore, the same periods are chosen for all missions, based on the Jason cycles. We actually did a lot of work that could not be included in the paper because it is a bit too methodological and tedious to explain, but we've shown that it is very important to compute grids from 2 missions on the same exact time periods, subtract those grids, and then compute GMSL. If you simply take 2 GMSL timeseries from different missions, you must interpolate on a common sampling, and you lose a lot of HF variation doing that, which affects our method by giving rise to an underestimation of the trend uncertainties. Directly subtracting 2 GMSL timeseries also does not ensure that the source data is matched spatially. So, using the method we described, we evaluated that 10 days is enough to have a global representation (Jason cycles) and keep enough of the HF variations.

Generally the paper was very clearly written, such that I only find a few minor errors worth mentioning.

Graham Quartly

Suggested corrections

l. 17 Possibly "maximum detectable" should be "minimum detectable"? Indeed. Corrected in the paper.

l. 41 Change 'data is' to 'data are'. Yes. Corrected in the paper.

l. 44 'C3S' should be expanded at first use. Yes. Corrected in the paper.

l. 88 Please expand on why data used for calibration purposes are discarded, as you do later use the calibration phase of S3B. Following the recommendations of S3A Cal/Val team, we have excluded  the beginning phase of S3A due to a lot of gaps and changes in instrument modes. Concerning S3B, we did not observe as many gaps in its calibration phase as for the S3A calibration phase. Since the period was short, we decided to use the maximum data available.

l.101 Useful to also cite Frery et al., 2020. Yes, thank you for bringing it to our attention.

Fig. 1 There are common short-term variations for red and grey curves suggesting that much of the short-term variability is due to AL. Is this due to it being in a drifting orbit and thus does not provide complete global coverage on monthly timescales?

Indeed, it appears that we can observe correlated short-term variations between the red (J3/AL) and gray (S3A/AL) curves suggesting that it originates from AL.  We did not investigate this short-term signal in this study because it is considered as a source of uncertainty in our error budget approach to estimate sea level drift uncertainties. In short, some of the short-term variations comes from the error between the measurements of the 2 altimeters, and some comes from the intrinsic variability of the ocean that is not observed in the same way by the different satellites. The fact that AL is on a drifting orbit could indeed contribute to a particular observation of the ocean variability but at this stage it is not possible to determine whether it comes from the drifting orbit and variation in the ocean sampling, or from a signal in one of the geophysical correction or range.

l. 172-175 Is it correct to assume that errors in SSB will be mainly sub-annual? For many regions the wave field has a strong annual signal (not just the Atlantic, but parts of the

Indian Ocean where winds will have significantly different fetches according to phase of monsoon). In the paper we look for all altimeter-induced drift, so all long term (> 1 year) errors of Range, SSB, or ionospheric correction, are precisely what we aim to detect. Therefore, we do not include it in the error budget, which only contains errors that are preventing us from measuring accurately the altimeter-induced drift.

In the future, it would be relevant to be able to better characterize the long-term errors of the SSB, in order to isolate the long-term errors coming only from the altimeter range. However to date, the characterization of the SSB errors is not mature enough to be taken into account in our error budget approach.

We have modified the text at the end of section 2 to make it clearer that we consider all altimeter-induced drift i.e. not only the range: 'However, in this study, we choose to focus on altimeter induced drift which affects the altimeter range and SSB, as well as the ionospheric correction.'

l. 217 Change 'splitted' to 'split'. Yes. Corrected in the paper.

l. 261 Should this be ''Aublanc, 2020'? Yes. Corrected in the paper.

l. 262 Delete first instance of 'correction'. Yes. Corrected in the paper.

l. 278 Should be 'S3B'. Yes. Corrected in the paper.

l. 280-281 Needs revising in light of Dinardo's findings. Yes, we updated this paragraph to say that those results were not available at the time of this study and briefly say that it was an inverted sign in the implementation of the USO correction that was the cause of this issue, and reference to Dinardo et al. 2021.

l. 292 I do not understand the point being made 'The knowledge of the statistical behaviour of the error is a difficult task.' Please reword or remove. Yes, you are right, this sentence is unnecessary. Corrected in the paper.

l. 296 Suggest replace 'this time' with 'instead'. Yes. Corrected in the paper.

l. 317 In the light of Dinardo's findings on error in USO correction, please comment here on whether the differences are still significant or now understood. As explained in lines 180-181, we have updated the paper referencing the finding from Dinardo et al. 2021 on error in USO correction. But we made the choice to not update the results with the correct application of the USO correction in this paper : it might be performed in further studies.

l. 376 Change 'march' to 'March'. Yes. Corrected in the paper.

l. 377 In the light of Dinardo's findings, please revise, comment or remove this sentence. Yes, like for lines 180-181, we've referenced Dinardo et al. 2021 and briefly state that it is due to an incorrect implementation of the USO correction.

l. 381 I think 'up to' should be replaced with 'over'. Yes. Corrected in the paper.

l. 394 Need to use superscript (twice). Yes. Corrected in the paper.

References: Details are missing for Ablain, 2018; Aublanc, 2020; Jettou amd Rousseau, 2020; Meyssignac, 2019; Roinard and Michaud, 2020. Also for citing OSTST presentations (i.e. Poisson, 2019), one should give an address where they can still be accessed and the date that you last did so. Yes. Corrected in the paper.

Frery, M.-L.; Siméon, M.; Goldstein, C.; Féménias, P.; Borde, F.; Houpert, A.; Olea Garcia, A. Sentinel-3 Microwave Radiometers: Instrument Description, Calibration and Geophysical Products Performances. Remote Sens. 2020, 12, 2590. https://doi.org/10.3390/rs12162590

Quartly, G.D.; Nencioli, F.; Raynal, M.; Bonnefond, P.; Nilo Garcia, P.; Garcia-Mondéjar, A.; Flores de la Cruz, A.; Crétaux, J.-F.; Taburet, N.; Frery, M.-L.; Cancet, M.; Muir, A.; Brockley, D.; McMillan, M.; Abdalla, S.; Fleury, S.; Cadier, E.; Gao, Q.; Escorihuela, M.J.; Roca, M.; Bergé-Nguyen, M.; Laurain, O.; Bruniquel, J.; Féménias, P.; Lucas, B. The Roles of the S3MPC: Monitoring, Validation and Evolution of Sentinel-3 Altimetry Observations. Remote Sens. 2020, 12, 1763. https://doi.org/10.3390/rs12111763

---

## Author Comment (AC4)

**Response to second reviewer (anonymous)**

Response: Thank you very much for your review, we have complied with all of the changes you've requested.

Kind regards,
Rémi Jugier

General: I appreciate having been given the opportunity to review a preprint of "On the uncertainty associated with detecting global and local mean sea level drifts on Sentinel-3A and Sentinel-3B altimetry missions" by Jugier et al. The paper is well organized and written, and provides an excellent overview of sources of uncertainty in satellite altimetry measurements of sea level trends, differences between missions, and implications for resulting sea-level rise estimates. It rigorously quantifies drift in GMSL trends between satellite altimetry missions and by comparison to tide gauge records, allowing for some of the uncertainty to be accounted for, and guiding future improvements in altimetry and uncertainty analysis. The work provides important new insight to uncertainty in satellite altimetry-based inferences of sea level, which is crucial for understanding future sea level changes and associated impacts, as well as the use of altimetry data to support modelling. I provide only some minor specific comments and suggested technical corrections.

Specific comments:

Lines 33-35: It is not immediately clear what is meant by "At the local scale", although the authors refer to a scale of 2400 km in Line 18. I suspect the authors are referring to the fact that there are spatial variations in sea-level rise across the global oceans. If so, I suggest changing this sentence to begin "Rates of sea-level rise vary spatially in the range 0 to 6 mm yr-1...". Alternatively, explain what is meant by local scales, e.g. "Over distances of 2400 km, sea-level rise rates vary by between 0 and 6 mm yr-1..." I note that later in the paper, three different local scales (240 km to 2400 km) are referred to. It is not clear which of these local scales the range 0-6mm/yr applies to. We agree with you that the use of the term "local scales'' can be confusing. In our minds, we used "local scales" as opposed to "global scale". In practice, "local scales'' range from a hundred kilometers to a few thousand kilometers. We have applied your first suggestion, and elsewhere in the paper, as you've suggested, we've made things clearer. Firstly, we've used "regional scales'", which resonates more with "global scale" and is probably clearer for the reader than "local scale". The term "regional scales" has also been explicitly defined when first used in the paper to avoid ambiguity. And then, we've implemented your suggestions of speaking

about "spatial variability of sea level drifts". We've also specified more explicitly what was the spatial scale each time that we were using "local scale".

Lines 73-74: Again, it is not quite clear what the authors mean by "local scales". I would suggest to change this to say something like "...we assess spatial variability in the drift in sea level estimates" As explained just before, we have improved this sentence to say that we extend the detection of the global sea level drift to different regional scales, i.e. assess spatial variability in the drift in sea level estimates.

Line 109: It is not clear why a resolution of 1 degree latitudinally and 3 degrees longitudinally are selected for this method. Could the authors clarify this choice of resolution, and comment on the potential influence of grid resolution and spatial collocation of altimetry tracks to the reference grid on the resulting analysis and MSL drift estimates? The resolution of 1 degree latitudinally and 3 degrees longitudinally applied in this study is coming from the GMSL AVISO calculation method (https://www.aviso.altimetry.fr/en/data/products/ocean-indicators-products/mean-sea-level/processing-and-corrections.html ). Henry et al. (2014) showed this resolution is the most adapted for the historical orbit of the TOPEX/Jason altimeter missions in order to calculate the GMSL time series and well represent its inter-annual variations (e.g. during ENSO events for instance) . We have decided in this study to keep this resolution to detect the relative GMSL drift between Sentinel-3 (A and B), SARAL-Altika, and Jason-3 first because we wanted to stay in line with the GMSL AVISO method. We have included more information in the updated paper to better justify the spatial resolution applied for this study.

Line 321: Clarify the reference time period used to determine trends in Section 4.4. Yes, added in the paper.

Lines 354-358: Do these findings suggest that some spatial averaging/smoothing is generally needed or recommended to obtain robust sea-level estimates? Indeed, spatial smoothing/averaging reduces the impact of ocean variability, which is the limiting factor for detecting relative sea level drift between these same two missions. As shown in the paper, taking into account large regional areas (e.g. 2400 km) reduces the uncertainties in the relative sea level trend. However, this smoothing/averaging approach is done at the expense of small spatial scales. Our recommendation would be to find a method where the effect of oceanic variability should be reduced (e.g. by removing as much as possible the mesoscale signal) before comparing sea level estimates between two missions.

Line 380: Clarify what is meant by local scales here. Corrected by removing "local scale" and directly giving the spatial scale to which the values correspond.

Technical corrections:

Line 9: "It could have an impact on sea level rise of a few tenths of mm yr-1." This seems to imply the drift impacts the actual sea levels, which of course is not the case. I suggest changing to something like: "It will affect the accuracy of sea level sensing, which could result in errors in sea-level change estimates of a few tenths of mm yr-1" Yes. Corrected in the paper.

Line 12: Global Mean Sea Level should be written in full here, since it is the first time the GMSL acronym is used. Yes. Corrected in the paper.

Line 25: Suggest to insert the word "lagging" before "indicator" in the statement "GMSL rise is a widely accepted indicator for the rate at which the climate is changing". Your comment suggests that the characteristic response time of the ocean to global warming is long enough for the GMSL rise to be defined as a "lagging Indicator of climate change". After discussing with Benoit Meyssignac, expert of this topic, "Lagging indicator of climate change" is not adapted because it suggests that physical climate change should be reduced to the forcing applied to the climate system but not to the response of the climate system which develops over thousands of years. Benoit Meyssignac has also suggested us to change this sentence by "The GMSL rise is a widely accepted indicator of the current climate state (Meyssignac et al., 2019) and the GMSL acceleration for the rate at which the climate is changing" which is more exact.

Lines 34-45: The statement "...sea level is rising everywhere over the globe" is not necessarily true depending on the reference frame and location. For example, at coastal locations experiencing post-glacial rebound (e.g. the Canadian High Arctic), sea levels are actually falling relative to the land. Some clarification is probably needed. Prandi et al, (2021) shows that 98% of the ocean surface experiences a significant sea level rise (Fig. 5 of this study). The few regions where sea level trends are not significant are located in the Southern Ocean, Baffin Bay and in the north Atlantic Ocean, south of Iceland. In all areas where sea level is falling (see Fig. 5) the rate of sea level fall is not significant at the 90% confidence level, except in the Caspian Sea.  We have modified the text in the paper adding "sea level is rising almost everywhere over the globe".

Line 56: Suggest to insert the word "inferred" before "GMSL" in the statement "...with a direct impact on the GMSL trend of about 0.3 mm yr-1."  Yes. Corrected in the paper.

Line 109: There appears to be too many ")" in this sentence. Yes. Corrected in the paper.

Lines 152 and 153: Insert space after GMSL (two instances) Yes. Corrected in the paper.

Table 1: GIA errors are canceled out Yes. Corrected in the paper.

Table 1 footnote: All uncertainties reported are based on Gaussian distributions Yes. Corrected in the paper.

Table 2: see comments on Table 1 – same apply here. Yes. Corrected in the paper.

Line 243: I think this line should be altered to state "These results highlight a significant difference in GMSL trends estimated from S3A and Jason-3…" Yes. Corrected in the paper.

Line 373: I did not find the cited Poisson et al. (2018) reference in the bibliography. Perhaps this should be Poisson et al. (2019)? Yes, it is 2019. Corrected in the paper.

Lines 409-468: Several of the references appear to be incomplete, lacking information needed to locate the publication. Yes, the other reviewer pointed this out as well. Those references are not articles but presentations or project reports. We've added more details and links to the presentations.

---

## Author Response (AR2)

**On the uncertainty associated with detecting global and local mean sea level drifts on Sentinel-3A and Sentinel-3B altimetry missions - Author response to reviewers (2nd round)**

**Response to first reviewer (Graham Quartly)**

I was a little disappointed that the authors did not use the opportunity offered by this resubmission to evaluate their approach using data consistent with what others will have by the time this is published i.e. with the correct USO correction for S3B. Fig. 4b of Quartly et al. (2020) shows the USO correction for S3B to change by 3mm over 2 years; given that this was originally applied with incorrect sign, one would infer a correct application would change the DelatGMSL by 3 mm/yr. A more thorough analysis could have been done by making appropriate use of the daily average correction

(ftp://ftp.eumetsat.int/pub/EUM/out/RSP/lucasb/S3B.USO/S3B.cor\_uso.daily.csv) to amend their 10-day gridded fields.

Response: We fully understand the point of view of the reviewer, however we would just like to remind that this study allowed us to detect and quantify the drift of the S3-B (and S3-A) GMSL in the framework of the S3-MPC project, and for this reason the USO correction was not yet available when the study was performed. For the sake of clarity, we believe it would be best to verify the stability of S3-B (and S3-A) corrected with this USO correction in a separate paper when the L2/L2P products are reprocessed and made available. If, however, this point is blocking the publication of the paper, a compromise may also be to provide this information in auxiliary material.

For me, the main plus of this paper is the robust statistical analysis to put uncertainty on the estimates of DeltaGMSL, and thus assign a probability that this is significantly different from zero. Indeed Figs. 2 & 3 could be usefully replaced with a table giving Mean trend, Uncertainty (1 sigma), Significance, rather than having some values in a figure and some in main text. I am not sure about their Eq. 4 -- can the authors please check that that is correct.

Response : Yes, you are right about the formula, which is now corrected to :

**$\hat{\beta} = N(\beta, (X^t X)^{-1}(X^t \Sigma X) (X^t X)^{-1})$**

For the figures vs table issue, we think that the figures make it easier to see the drifts and associated uncertainty, and whether they are significant or not by just looking at the intersection with 0. They also provide a means to quickly analyze the whole picture of missions intercomparisons, which is necessary to attribute the observed drifts to a specific mission for instance. We indeed provided trends, uncertainties, and using a gaussian distribution we calculated the confidence interval associated with trend / uncertainty, but this last statistic is only provided in the text. If the editor thinks it is necessary to add a table to sum up those statistics, we could add a table in the conclusion.

Seven of the given references are "grey literature" i.e. conference presentations or unrefereed reports. These should all have date when last accessed. Presuming Legeais et al. 2021 is published it should have volume and page no. Henry reference should start on a new line.

Response : OK, we've checked that the access was still working for the grey litterature references where we provided the link, and added the current date. The links to the presentations were also added for :

Ablain, M.: Estimating of Any Altimeter Mean Sea Level (MSL) drifts between 1993 and 2017 by Comparison with Tide-Gauges Measurements, 25 years of progress in altimetry radar symposium, 2018. https://drive.google.com/file/d/1Wt7nDLBOwtjGYtDPoO1ofFoisyuFv1yu/view?usp=sharing (last accessed June 10th 2022).

Poisson, J. C., Piras, F., Raynal, M., Cadier, E., Thibaut, P., Boy, F., Picot, N., Borde, F., Féménias, P., Dinardo, S., Recchia, L., and Scagliola, M.: SENTINEL-3A instrumental drift and its impacts on geophysical estimates, OSTST 2019, 2019.

https://ostst.aviso.altimetry.fr/fileadmin/user\_upload/2019/IPM\_02\_Poisson\_OSTST2019\_PTR\_Drift.pdf (last accessed June 10th 2022).

In the case of Abain et al., it seems like the 25 years of progress in altimetry radar symposium, 2018 conference did not make presentations accessible through public links, so we put the presentation onto a google drive and generated the link for readers.

Legeais et al. 2021 was indeed published, but google scholar citations are not up to date, so I've added volume, page number, and DOI manually.

Henry et al. reference was indeed moved to a new line.

The paper details quite a bit the procedure and analysis for regional variation in DeltaGMSL and ends up showing that no region is appreciably different from the global value. Given that the authors have quite correctly minimised differences in corrections by using consistent atmospheric and tide models throughout, it would be helpful if they devoted a few sentences to which mechanisms could be responsible for regional variations and thus what this null result allows us to regard as understood.

Response: We have indeed detected a significant drift in the GMSL of S3A/S3B with respect to the Jason-3, Jason-2 SARAL/Altika mission, which we attributed to the altimetric parameters of these 2 missions (e.g. altimetric range, SWH). We performed the same type of analysis at regional scales to verify if this global drift could have a regional signature. At regional scales, the main sources of error can come from inhomogeneity in the calculation of the orbit between 2 missions when 2 satellites are not at the same altitude, for example because of the modeling of the gravity fields at the ITRF (see Couhert et al.; 2015; Prandi et al., 2021). If instrumental drifts of the altimeter are present, one could also expect to observe a spatial signature that depends on the wave amplitude for example. Given the larger uncertainties of the method at these regional scales, we did not indeed detect any regional drift around the observed global mean value of the S3A/S3B GMSL drive.

In order to follow your recommendation, we have specified in the introduction the regional signals that we could possibly observe.

Minor corrections Abstract I.5 : possibly expand S3A and S3B on first use. Abstract I.6-7 : Drift of S3A is compared with J3 then AL; whereas S3B is compared with AL then J3. Please reverse 2nd pair to make them consistent. Introduction I.18 GSML should be GMSL Introduction I.21 Suggest change "lower revisit rate" to "longer revisit period" Introduction I.32 "Jason-3" occurs twice. Introduction I.39 "GMSL.s" is a little ugly; clearer would be "GMSL estimates" Section 3.1 I.34 Need to note that the 58.77-day alias is for Jason-3. Section 4.4 I.6 "from -2 to +2" Conclusions I.12 S3B is compared with AL then J3. Please reverse this pair to make them consistent with earlier S3A comparison Section 3.1 I.27-28 This states that time correlated errors can be divided into those with timescales < 1 yr and those for 5-10 years. What about 1-5 years, why are those so explicitly ignored?

Response: In fact, two types of errors affect our approach: a) error due to measurements, and b) error due to internal ocean variability observed differently by two satellites that are not in the same orbit.

a) For the error due to measurements, as discussed in the paper, our study is based on Ablain et al. 2019 where the main sources of errors impacting the stability of the GMSL have already been identified. According to Ablain et al. 2019, the time-correlated errors were divided at the same time scales: 2 months and 1 year, 5 years , 10 years, etc... Indeed, there are no descriptions in the literature of significant correlated altimeter errors at time scales between 1 and 5 years.

b) for the error due to internal ocean variability observed differently by two satellites is mainly observed for scales smaller than a few months (e.g. mesoscale) using and adapted averaging of the SLA grids ( according to Henry et al. (2012))

We've modified this paragraph to make it clearer.